# Dysregulation of Glucocorticoid Receptor Homeostasis and Glucocorticoid-Associated Genes in Umbilical Cord Endothelial Cells of Diet-Induced Obese Pregnant Sheep

**DOI:** 10.3390/ijms25042311

**Published:** 2024-02-15

**Authors:** Eugenia Mata-Greenwood, Wendy L. Chow, Nana A. O. Anti, LeeAnna D. Sands, Olayemi Adeoye, Stephen P. Ford, Peter W. Nathanielsz

**Affiliations:** 1Lawrence D. Longo Center for Perinatal Biology, School of Medicine, Loma Linda University, Loma Linda, CA 92350, USA; wendychow001@gmail.com (W.L.C.); nanti@students.llu.edu (N.A.O.A.); leeannasands@gmail.com (L.D.S.); 2Department of Pharmaceutical Sciences, School of Pharmacy, Loma Linda University, Loma Linda, CA 92350, USA; oadeoye@llu.edu; 3Center for the Study of Fetal Programming, Department of Animal Science, University of Wyoming, Laramie, WY 82071, USApnathanielsz@uwyo.edu (P.W.N.)

**Keywords:** maternal obesity, fetal programming, endothelial cell, glucocorticoid

## Abstract

Maternal obesity (MO) is associated with offspring cardiometabolic diseases that are hypothesized to be partly mediated by glucocorticoids. Therefore, we aimed to study fetal endothelial glucocorticoid sensitivity in an ovine model of MO. Rambouillet/Columbia ewes were fed either 100% (control) or 150% (MO) National Research Council recommendations from 60 d before mating until near-term (135 days gestation). Sheep umbilical vein and artery endothelial cells (ShUVECs and ShUAECs) were used to study glucocorticoid receptor (GR) expression and function in vitro. Dexamethasone dose–response studies of gene expression, activation of a glucocorticoid response element (GRE)-dependent luciferase reporter vector, and cytosolic/nuclear GR translocation were used to assess GR homeostasis. MO significantly increased basal GR protein levels in both ShUVECs and ShUAECs. Increased GR protein levels did not result in increased dexamethasone sensitivity in the regulation of key endothelial gene expression such as endothelial nitric oxide synthase, plasminogen activator inhibitor 1, vascular endothelial growth factor, or intercellular adhesion molecule 1. In ShUVECs, MO increased GRE-dependent transactivation and FKBP prolyl isomerase 5 (FKBP5) expression. ShUAECs showed generalized glucocorticoid resistance in both dietary groups. Finally, we found that ShUVECs were less sensitive to dexamethasone-induced activation of GR than human umbilical vein endothelial cells (HUVECs). These findings suggest that MO-mediated effects in the offspring endothelium could be further mediated by dysregulation of GR homeostasis in humans as compared with sheep.

## 1. Introduction

Maternal obesity (MO) during pregnancy occurs in ~30% of human pregnancies and is an important risk factor for maternal and fetal complications [1,2]. In addition, cumulative human and animal evidence shows a definitive link between MO and the offspring’s postnatal risk of disease [3,4,5,6,7]. Epidemiological and in vivo research studies have shown that MO offspring have a higher risk of cardiovascular disease because of a rise in risk factors such as obesity, endothelial dysfunction, atherosclerosis, thrombosis, and hypertension [8,9,10]. However, the effects and mechanisms of MO on the offspring’s cardiovascular system remain incompletely understood.

It has been suggested that the MO programming of postnatal cardiovascular disease in offspring is partly due to hormonal and inflammatory mediators that alter vascular tissue homeostasis [8,9,10]. One of these mediators is cortisol, the main endogenous glucocorticoid that regulates energy homeostasis and stress-coping responses [10,11,12]. MO can increase maternal, fetal, postnatal, and intergenerational cortisol levels and reset the threshold and regulation of the hypothalamic–pituitary–adrenal (HPA) axis [13,14,15,16]. In vivo studies have also demonstrated that MO increases glucocorticoid receptor (GR) expression in the offspring heart tissue at fetal and adult stages, in association with hypertension and heart extracellular matrix remodeling [17,18]. In addition, supraphysiological cortisol levels observed in Cushing’s syndrome or chronic treatment with systemic synthetic glucocorticoids result in metabolic and cardiovascular effects similar to those induced by obesity [19,20,21]. Therefore, MO dysregulation of glucocorticoid homeostasis is a potential mechanism of the downstream effects on offspring health. 

Glucocorticoid’s effects on the cardiovascular system are the result of the combinatorial changes in myocardial, endothelial, smooth muscle, fibroblast, and immune cell phenotype. In endothelial cells, cortisol or synthetic glucocorticoids can have beneficial anti-inflammatory and anti-atherosclerotic effects in blood vessels by the downregulation of pro-inflammatory molecules such as intercellular adhesion molecule 1 (ICAM1) [21,22,23]. However, in the setting of obesity or chronic inflammatory disease, glucocorticoid’s anti-inflammatory effects are decreased, and pro-atherogenic and vasoconstrictive effects are increased [21,22,23]. Proposed molecular mechanisms of glucocorticoid-induced hypertension and atherosclerosis in endothelial cells include downregulation of endothelial nitric oxide synthase (NOS3) and vascular endothelial growth factor (VEGFA) and upregulation of endothelin-1, factor VIII, and plasminogen activator inhibitor 1 (SERPINE1) [21,22,23].

Tissue glucocorticoid sensitivity is mainly determined by the levels and function of GR [24]. The regulation of GR expression and function is complex and highly sensitive to environmental, metabolic, and epigenetic regulation [24,25,26,27]. Unliganded GRα, the most active isoform, resides in the cytosol bound to chaperones, of which FKBP prolyl isomerase 5 (FKBP5) is an important negative feedback regulator that decreases ligand affinity and GRα nuclear translocation [28,29]. Once activated by ligand binding, GRα translocates to the nucleus and regulates the transcription of multiple genes in a cell-type- and environmental-specific manner [24,27]. A classic example of GRα transcription regulation is transactivation by GRα homodimers via interaction with glucocorticoid response element (GRE) domains of target genes such as FKBP5 [24,30]. Therefore, glucocorticoid activity leads to upregulation of the endogenous GR inhibitor FKBP5, providing an ultrashort negative feedback mechanism. Monomeric GRα also regulates the transcription of multiple genes by various mechanisms such as negative GRE-mediated transrepression and DNA-free tethering with other transcription factors [31,32]. 

The aim of this study was to determine the effect of MO on fetal endothelial cell glucocorticoid sensitivity. We hypothesized that MO would increase the expression of GR and glucocorticoid sensitivity, resulting in a deleterious endothelial phenotype characteristic of endothelial dysfunction. We observed that MO upregulates fetal endothelial cell GR but decreases its activation. We also found that sheep endothelial cells are relatively resistant to glucocorticoids compared with human endothelial cells.

## 2. Results

### 2.1. Maternal Obesity Upregulates GR Expression in ShUVECs and ShUAECs

The ovine model of MO used in the current study has been shown to exhibit higher maternal and fetal cortisol plasma levels and higher myocardial GR expression [16,17,33]. In this study, the obesogenic diet significantly increased maternal weight gain but not birthweight (Table 1). Studies on ShUVECs and ShUAECs revealed that maternal overnutrition decreased basal GRα mRNA levels in ShUVECs by 41% and increased basal GR protein levels in ShUVECs by 43% and in ShUAECs by 29% (Figure 1). In addition, there were no significant differences in basal GR expression between ShUVECs and ShUAECs (Figure 1). Although glucocorticoids are well known to induce GR protein proteasomal degradation [34], dexamethasone had no significant effect on GR protein levels in ShUVECs (Figure 1A–C), but it significantly decreased GR protein levels in ShUAECs with a maximal decrease of 44.1% in CO-ShUAECs and 33.9% in MO-ShUAECs (Figure 1E,F).

### 2.2. Maternal Obesity Dysregulates Key Endothelial Gene Expression

Next, we investigated the effect of MO on the expression of key endothelial genes known to be regulated by cortisol and synthetic glucocorticoids. We found that MO decreased NOS3 expression in ShUVECs by 63-145% (Figure 2A) and increased SERPINE1 expression in ShUAECs by 67-146% (Figure 2B). ShUVECs expressed higher levels of NOS3 and ICAM1 but lower levels of VEGF and SERPINE1 than ShUAECs (Figure 2A–D). Interestingly, both ShUAECs and ShUVECs were resistant to dexamethasone-mediated transcriptional regulation of NOS3, SERPINE1, VEGFA, and ICAM1 (Figure 2A–D), although there was a non-significant dexamethasone-mediated upregulation of SERPINE1 in MO-ShUAECs (Figure 2B).

### 2.3. Maternal Obesity Dysregulates GR-Homeostasis

To uncover the role of MO on endothelial GR function, we studied the transcriptional activation of a consensus GRE-driven luciferase reporter vector. We found that in ShUVECs, MO decreased basal and dexamethasone-stimulated GRE-driven transcription by more than 2-fold (Figure 3A). In contrast, MO did not alter basal or dexamethasone-induced GRE-dependent transcription in ShUAECs (Figure 3B). In addition, dexamethasone induced a dose-dependent transactivation of GRE-dependent luciferase expression in ShUVECs of both dietary groups with a maximal increase of 70% in CO-ShUVECs and 95% in MO-ShUVECs (Figure 3A). In contrast, dexamethasone did not significantly change the GRE-driven transcriptional activation of ShUAECs with the exception of a significant 53% increase in MO-ShUAECs at the highest dose of dexamethasone (Figure 3B). 

We then investigated nuclear-to-cytosolic GR ratios in basal and dexamethasone-stimulated cells as only nuclear GR can act as a transcription factor. MO decreased nuclear/cytosolic GR ratios in MO ShUVECs by 45% in solvent-, 52% in dexamethasone low-dose-, and 68% in dexamethasone high-dose-treated cells (Figure 4A,B). Dexamethasone at low and high concentrations increased GR nuclear/cytosolic ratios in CO-ShUVECs, while only high-dose dexamethasone increased nuclear GR in MO-ShUVECs (Figure 4A,B). Importantly, the GR nuclear/cytosolic ratios in ShUVECs were correlated with GRE-mediated transactivation of a luciferase reporter (Figure 3A and Figure 4A,B). In contrast, the MO diet did not affect basal GR nuclear translocation but decreased dexamethasone-induced nuclear translocation by 38–39% in ShUAECs (Figure 4B). Furthermore, dexamethasone-induced GR nuclear translocation in CO- and MO-ShUAECs was higher in ShUAECs compared with ShUVECs (6-fold increase in CO-ShUAECs, 3-fold increase in MO-ShUAECs, 2-fold increase in CO-ShUVECs, and 1.75-fold increase in MO-ShUVECs; Figure 4A–D). Interestingly, the dexamethasone-induced increases in GR nuclear translocation in ShUAECs did not result in increased GRE-transcriptional activation as shown by luciferase assays (Figure 4C,D vs. Figure 3B), suggesting additional nuclear inhibitors of GR function in ShUAECs.

### 2.4. Maternal Obesity Increases FKBP51 Upregulation as a Negative Feedback Mechanism

GR is regulated by complex mechanisms, of which GR-dependent upregulation of FKBP5 is a classic negative mechanism that prevents glucocorticoid overshooting effects. The FKBP5 promoter contains GREs that render this gene highly sensitive to glucocorticoid transactivation, making it a hallmark of the glucocorticoid response [28,29,30]. FKBP5 mRNA and protein levels were upregulated by dexamethasone in a dose-dependent manner in MO-ShUVECs only (Figure 5A,B), with a maximum mRNA and protein increase of 76% and 91%, respectively, at the highest dose of dexamethasone. Although there were no significant differences in ShUAECs, there was a non-significant increase in MO-ShUAEC FKBP5 mRNA and protein compared with CO-ShUAECs and non-significant differences of 30–101% higher FKBP5 protein levels in ShUAECs compared with ShUVECs (Figure 5C,D).

### 2.5. Decreased Dexamethasone Sensitivity in ShUVECs Compared with HUVECs

Our previous studies demonstrated that HUVECs were sensitive to dexamethasone-mediated regulation of gene expression [35] in contrast with ShUVECs of both dietary groups (Figure 2). To confirm differential GR regulation in ShUVECs compared with HUVECs, we studied ShUVECs from another ovine breed together with previously characterized HUVECs [35,36]. Western Mix (WM) ShUVECs had a similarly high basal nuclear GRE-dependent transactivation but were resistant to dexamethasone-mediated further increases compared with CO-ShUVECs (Figure 6A vs. Figure 3A). Strikingly, ShUVECs from both breeds showed a higher basal GRE-dependent transcriptional activation that was ~4-fold higher than HUVECs (basal Firefly/Renilla luciferase ratio of 27 in ShUVECs and 6.5 in HUVECs, Figure 6A). In addition, while ShUVECs showed only a non-significant dexamethasone activation of GRE-driven transcription of 52% in WM-ShUVECs (Figure 6A) and 70% in CO-ShUVECs (Figure 3A), HUVECs showed a strong dose–response to dexamethasone with a maximal effect of nearly 7-fold (600% increase in basal levels, Figure 6A). These differences were associated with the subcellular localization of GR: WM-ShUVECs, similar to CO-ShUVECs, showed higher basal nuclear GR residence and smaller increases by dexamethasone stimulation compared with HUVECs (Figure 4A and Figure 6B). However, WM-ShUVECs showed a lower basal GR nuclear/cytosolic than CO-ShUVECs (0.72 in WM-ShUVECs, Figure 6B, vs. 1.89 in CO-ShUVECs, Figure 4A). In addition, lipopolysaccharide (LPS) increased GR and RELA nuclear translocation in WM-ShUVECs but had no effect in HUVECs (Figure 6B,C). Altogether, HUVECs have lower basal levels of nuclear GR, resulting in a lower basal activation of GRE-dependent transcription that allows for a higher response to dexamethasone activation, as shown by the ~7-fold increase in GRE-mediated transcriptional activation and GR nuclear translocation (Figure 6A,B). Changes in HUVEC-GR protein nuclear translocation then associate with glucocorticoid sensitivity in terms of dexamethasone-regulation of endothelial gene expression [35]. In contrast, ShUVECs show endogenous resistance to glucocorticoid activation of GR and transcriptional regulation of key genes.

### 2.6. MO Increases NF-kB Expression in Umbilical Endothelial Cells

Because WM-ShUVECs showed higher basal and stimulated nuclear NF-kB RELA levels (Figure 6C), we investigated the effect of MO on total and nuclear RELA levels (Figure 7) in all four groups. We found that MO increased the basal levels of total p65 NF-kB in ShUVECs by 87% (Figure 7A,B) and ShUAECs by 76% (Figure 7C,D) without increasing nuclear translocation (Figure 7E,F). Indeed, more than 90% of RELA protein was localized in the cytosol of all samples (Figure 7E,F). Furthermore, dexamethasone did not have any effect on RELA expression or subcellular location, although there was a trend towards RELA downregulation in MO-derived cells (Figure 7A,C).

## 3. Discussion

### 3.1. Maternal Obesity Effects on Fetal Endothelial GR Expression and Function

In this study, we uncovered novel effects and potential mechanisms of MO-induced dysregulation of fetal endothelial GR homeostasis. We confirmed our hypothesis that maternal obesity increases GR expression in fetal umbilical artery and vein endothelial cells. Previous studies using this ovine model have observed increased cardiac ventricular GR mRNA/protein expression in adult offspring born of obese ewes in correlation with increased plasma cortisol levels, thickening of both right and left ventricular walls, increased cross-linking of myocardial collagen, and tissue expression of pro-inflammatory mediators such as TNF-alpha [17]. A previous study revealed that MO effects in offspring myocardium are evident as early as 75 days of gestation and manifested as increased wall thickness and tissue expression of pro-inflammatory markers such as interleukin 6 (IL6) [18]. Therefore, MO-upregulation of fetal endothelial cell GR expression in the current study could be mediated by inflammatory mediators since reports have shown that IL6 and LPS upregulate GR expression in other tissues [37,38]. Furthermore, MO upregulation of fetal endothelial GR protein expression did not correlate with mRNA levels, suggesting translational and post-translational mechanisms of GR regulation, such as decreased microRNA interference with translational processes [39] or post-translational modifications that decrease protein degradation [34]. Additional studies are required to uncover MO-related regulators of endothelial cell GR expression.

Perhaps the most interesting finding of this study was that increased GR protein levels did not translate into higher dexamethasone sensitivity, as determined by a battery of assays that include the ability to downregulate its own receptor, transactivate a GRE-luciferase reporter vector, and regulate transcription of key genes. In ShUVECs of both dietary groups, dexamethasone treatment did not decrease GR protein levels or regulate transcription of NOS3, SERPINE1, VEGF, and ICAM1. These results contrast with those obtained with a cohort of HUVECs, where dexamethasone reduced GR protein levels to a fourth of basal levels, decreased NOS3 and VEGFA, and increased SERPINE1 and ICAM1 expression [35]. However, ShUVECs of both dietary groups were partially sensitive to dexamethasone, as shown by the increased GRE-dependent transactivation of a luciferase reporter vector and stimulation of GR nuclear translocation. In addition, MO-ShUVECs were sensitive to dexamethasone upregulation of the GR chaperone FKBP5, known to decrease ligand binding affinity and prevent nuclear localization [28,29]. These data suggest that increased GR protein levels in MO-ShUVECs are a result of increased cytosolic GR that is retained by association with FKBP5, thereby preventing nuclear GR translocation and transcriptional function. This would explain why MO did not increase overall glucocorticoid sensitivity in ShUVECs (summarized in Figure 8).

ShUAECs, on the other hand, showed a different response to dexamethasone compared with ShUVECs. Dexamethasone increased GR nuclear localization at higher levels than those observed in ShUVECs, yet these effects did not correlate with increased GRE-dependent luciferase transcription or dexamethasone transcriptional regulation of NOS3, SERPINE1, VEGF, and ICAM1. Furthermore, ShUAECs were resistant to dexamethasone upregulation of FKBP5 expression. These data suggest that ShUAECs’ resistance to dexamethasone could be due to nuclear GR protein turnover by proteasomal degradation and other nuclear-specific inhibitory elements that limit GR function as a transcription factor (summarized in Figure 8). Altogether, MO effects varied between ShUVECs and ShUAECs; the former showed only partial sensitivity to dexamethasone, as shown by stimulation of GR nuclear translocation, GRE-luciferase transactivation, and FKBP5 upregulation, which ultimately prevented further transcriptional regulation. ShUAECs of both dietary groups were more resistant to dexamethasone than ShUVECs, as shown by the inability to transactivate a GRE-luciferase reporter vector and upregulate FKBP5 expression. We hypothesize that generalized resistance to dexamethasone-mediated transcription regulation in ShUAECs is due to epigenetic (i.e., methylation) and cell-specific nuclear inhibition of GR. A decreased response of fetal endothelial cells to glucocorticoids could represent a mechanism to protect the vasculature from glucocorticoid-mediated endothelial dysfunction.

### 3.2. Maternal Obesity Effects on Key Endothelial Gene Expression

The current study revealed that MO altered the expression of key endothelial genes according to the endothelial cell source. In particular, MO induced downregulation of NOS3 in ShUVECs and upregulation of SERPINE1 in ShUAECs. Decreased NOS3 expression could lead to lower nitric oxide (NO) bioavailability, a hallmark of endothelial dysfunction [40], while increased PAI-1 could stimulate atherosclerosis and clot stability (Figure 8). Human studies have observed that adult obesity correlates with decreased NO bioavailability, increased SERPINE1 levels, and microvascular endothelial dysfunction shown as decreased Doppler flow curves. A recent study in HUVECs revealed that maternal obesity was associated with decreased placental transport of L-arginine, a substrate of NO [41]. Another mechanism of obesity-mediated decreased NO bioavailability is increased reactive oxygen species production that quenches NO [42]. Of interest, most studies have focused on the regulation of NOS3 in arteries due to the biological effects of NO as a vasodilating, platelet inhibitor, and antiatherosclerosis agent; yet, NO has important functions in other vessels such as systemic veins and pulmonary vasculature [43]. Therefore, MO-induced downregulation of vein endothelial cell NOS3, if correlated with reduced NO production and bioavailability, could be an important risk factor for the development of embolism or deep vein thrombosis [44], which are recurrent and common vascular diseases in obese patients. Another key molecule mediating glucocorticoid-deleterious effects in the endothelium is SERPINE1, a molecule that participates in the development of atherosclerosis and thrombosis and is thereby considered a risk factor for cardiovascular events such as myocardial infarction [45]. Of importance, studies have shown that maternal obesity is associated with increased maternal circulating SERPINE1 levels and endothelial dysfunction [46]. Endothelial cells and adipocytes are the main sources of plasma SERPINE1, and it has been shown that obesity upregulates its expression in both sources [47]. Therefore, MO programming effects on the offspring vascular system could include dysregulation of NOS3 and SERPINE1 expression. The differential responses to MO between ShUAECs and ShUVECs point to differential epigenetic and developmental regulation of endothelial gene expression and highlight the importance of studying vessels of different sources according to the disease of interest.

MO did not impact ICAM1 or VEGFA mRNA levels in ShUAECs or ShUVECs, although there was a tendency for MO-ShUVECs to respond to dexamethasone-mediated ICAM1 downregulation. Downregulation of ICAM1 expression is potentially highly significant in certain diseases considering the protective role of ICAM1 deficiency in sepsis-related mortality [48]. Indeed, human epidemiological studies have observed paradoxically that adult obese patients with sepsis show reduced mortality than non-obese and underweight patients [49,50]. Obesity-mediated upregulation of GR that results in increased cortisol sensitivity to ICAM1 downregulation could potentially explain these paradoxical clinical scenarios. Future studies on MO effects on the offspring response to infection or inflammatory stressors are therefore warranted. 

### 3.3. Comparative Endothelial GR Physiology: Human versus Sheep

A strikingly high level of basal nuclear GR was observed in ShUVECs in comparison with HUVECs. ShUVECs were grown in culture media containing insufficient amounts of cortisol to fully activate GR (~0.01 nM with cortisol having a Kd of 10 nM) and induce nuclear translocation in the absence of dexamethasone. Therefore, this is an intriguing novel observation. Hypothetically, we considered two scenarios: (1) differential GR gene structure that results in altered ligand affinity and/or (2) differential GR regulation by nuclear proteins in sheep compared with humans. Another key finding was the differential sensitivity of ShUVECs and HUVECs to LPS. ShUVECs were sensitive to LPS activation of NF-kB, as shown by nuclear translocation of RELA in contrast with HUVECs that showed resistance to low-dose LPS. HUVECs have been reported to have a decreased response to the pro-inflammatory effects of LPS, partly due to internalization of the toll-like receptor 4 [51]. Furthermore, we found that maternal obesity increased NFkB RELA levels in both ShUVECs and ShUAECs, suggesting an alternative pathway for fetal programming of endothelial dysfunction via pro-inflammatory mediators. Altogether, these data suggest that cortisol and GR homeostasis could have a stronger role in maternal obesity-programming effects in human offspring compared with that observed in sheep.

### 3.4. Conclusions

This study uncovered novel effects of MO on fetal umbilical endothelial cells that vary between arteries and veins. MO increased total GR protein levels but decreased nuclear GR presence, and in ShUVECs, increased cytosolic GR retention correlated with dexamethasone upregulation of the GR inhibitor FKBP5. ShUVECs showed higher basal nuclear GR presence with decreased responses to dexamethasone compared with HUVECs. We hypothesize that endothelial glucocorticoid resistance is a beneficial adaptation in sheep to prevent cortisol-induced endothelial dysfunction. Identifying the molecular mechanisms that lead to physiological resistance to glucocorticoid effects in sheep endothelial cells could translate into novel strategies to prevent glucocorticoid-mediated endothelial dysfunction in humans. The strengths of this study include parallel analysis of ShUAECs and ShUVECs from a well-characterized ovine model of maternal obesity, confirmatory studies using a different sheep breed, and comparative effects with HUVECs. Although studying umbilical vessel endothelial cells instead of other systemic vessels could be viewed as a weakness, the results obtained from this study can be compared with future studies on HUAECs and HUVECs from lean and obese pregnancies, which is an advantage. An important limitation is the small number of samples per group, which did not allow us to uncover the role of fetal sex in MO-mediated effects on endothelial phenotype. Lastly, cell-based functional studies such as cell proliferation, migration, angiogenesis, and wound healing in a pro-inflammatory setting are needed to fully uncover MO effects on the offspring endothelial phenotype.

## 4. Materials and Methods

### 4.1. Animal Care and Use

The animals and procedures used in this study were approved by the University of Wyoming and Loma Linda University Animal Care and Use Committees. Multiparous Rambouillet/Columbia cross ewes (with 2–3 previous pregnancies) were assigned randomly to a control (CO, 100% of National Research Council (NRC) recommendations) or obesogenic (MO, 150% of NRC recommendations) diet from 60 days before conception until the time of necropsy near term (137–140 days gestation with term = 145 days). Western mix ewes euthanized near term (139–141 days gestation) were used for confirmatory studies. Ewes were anesthetized with pentobarbital and euthanized by exsanguination for the collection of tissues including umbilical cords. 

### 4.2. Umbilical Cord Endothelial Cell Isolation

Approximately 3 inches of the umbilical cord, containing 2 arteries and 2 veins, were used to isolate sheep umbilical vein endothelial cells (ShUVECs) and artery endothelial cells (ShUAECs). The vessels were carefully separated from the Wharton Jelly, and the lumen was cleaned by flushing with sterile PBS before digestion with 1% collagenase II (Gibco, catalog 17101-015) in RPMI, as previously described [52]. Endothelial sheets were flushed with full media (Sciencell, catalog 1001) and allowed to grow to confluency in culture-treated plates and passaged using Trypsin-EDTA 0.25%. Cell culture purity was characterized by endothelial cell cobblestone morphology and immunocytochemical markers PECAM1 (Bio-Rad catalog MCA1097GA) and von Willebrand Factor (Abcam catalog ab6994). The first assays were performed between passages 1 and 3 and confirmatory assays between passages 4 and 6. 

### 4.3. Cell Culture, Treatment, and Determination of Glucocorticoid Sensitivity

To study the effect of an obesogenic diet on glucocorticoid sensitivity, we used dexamethasone (DEX) at 0.04, 0.2, and 1 µM concentrations using DMSO as solvent. This synthetic glucocorticoid was chosen for its stability in the presence of cellular 11-beta-dehydrogenase, and its specificity for GR with respect to other nuclear receptors such as the mineralocorticoid receptor. We also used lipopolysaccharide (LPS, O55:B5, Sigma Millipore, Burlington, MA, USA) at a low-dose concentration of 100 ng/mL. ShUVECs and ShUAECs were studied at confluency and quiescence using M199 minimal media (Sigma Millipore, catalog) supplemented with 0.95 mM HEPES, 0.1% BSA, 1% antibiotics, and 1% FBS. Glucocorticoid sensitivity was defined as a dose-dependent response change in endothelial gene expression and the ability to transactivate a GRE-dependent luciferase reporter assay [53,54].

### 4.4. Protein Isolation, SDS-PAGE, and Immunoblotting

Protein extracts were prepared in cold RIPA lysis buffer, or, alternatively, cytosolic and nuclear extracts were prepared using the commercial kit NE-PER (Thermo Fisher Scientific, catalog PI 78833, Waltham, MA, USA). Protein samples were heat-denatured in Laemmli buffer, separated on sodium dodecyl sulfate–polyacrylamide gel (SDS-PAGE), and transferred to polyvinylidene fluoride membranes. Membranes were blocked in 5% non-fat dried milk in 0.05% Tris-buffered saline (TBST) for 1 h and then probed in primary antibody overnight at 4 °C. The antibodies used are shown in Table 2. To determine the relative abundance of proteins, an internal control (pooled sample) was used in every membrane. All the antibodies were diluted in blocking buffer containing 5% non-fat dry milk in TBST. After three 10 min washes with TBST, the membranes were incubated with corresponding secondary antibodies that were diluted at 1:2000. Bound antibodies were visualized using the ChemiGlow Chemiluminescent substrate (Alpha Innotech Corp, San Leandro, CA, USA). Digital images were captured using the Alpha Innotech ChemiImager Imaging System with a high-resolution charge-coupled device camera and quantified using Alpha Innotech ChemiImager 4400 software. Equal loading was determined by beta-actin for total protein, HSP70 for cytosolic protein, and TBP for nuclear protein, and the relative abundance across multiple membranes was determined with the internal control, as previously described [35]. Results are shown as fold expression differences with respect to DMSO-treated CO-ShUVECs.

### 4.5. RNA Extraction, cDNA Synthesis, and Real-Time PCR

Total RNA was extracted with TriZOL (Invitrogen, Carlsbad, CA, USA), quantified, and stored at −80 °C until analysis. Total RNA (1 µg) was reverse transcribed using the Quantitect reverse transcription kit (Qiagen, San Diego, CA, USA). PCR reactions were performed in duplicate with Quantitect SYBR green qPCR kit (Qiagen, San Diego, CA, USA) using approximately 50 ng of total cDNA equivalent per reaction. PCR was run with denaturation at 95 °C for 15 s, annealing at 51–55 °C for 20 s, and extension at 72 °C for 10 s/100 bp. The BioRad iCycler equipped with a real-time optical fluorescent detection system was used for SYBR Green detection. β-Actin was found to be a more stable housekeeping gene than 18S and RPL19 [55]. Primers were designed using the NCBI primer design tool and tested for efficiency; only primers with efficiency between 90 and 110% were selected. The primer sequences, together with their accession numbers, are shown in Table 3. Negative controls (no template) and positive controls (reference sample) were included. To obtain relative fold mRNA levels we used the delta/delta Ct method and DMSO-treated CO-ShUVECs as the control group.

### 4.6. Cell Transfection and Lucierase Reporter Assays

Transfection of luciferase vectors was performed with the aid of HD Xtreme transfection reagent (Sigma Millipore). Luciferase assays were performed as previously described [35]. Briefly, confluent cells were transfected with the consensus glucocorticoid response element (GRE x2)-luciferase construct (Clontech, Palo Alto, CA, USA) using a 1:3 complex of DNA:transfection reagent and according to the manufacturer’s protocol. TK-Renilla luciferase vector (Promega Corp, Rockford, IL, USA) was used as the internal control in a 1:10 ratio with respect to the GRE-firefly luciferase vector. The transfection was carried out at 37 °C for 3 h. The cells were allowed to recover in complete culture medium for 12–16 h and were then treated with starvation media with or without dexamethasone for another 24 h. Firefly and Renilla luciferase activities were measured using a dual-reporter luciferase assay kit (Promega, Madison, WI, USA) according to the manufacturer’s protocol. Relative luciferase values were calculated as a ratio of Firefly/Renilla luciferase activities. Each treatment was tested in triplicates.

### 4.7. Statistical Analysis

Data are presented as means ± standard error. Differences between two groups (i.e., Control versus Obesogenic diet) were analyzed by Student’s unpaired *t*-tests. Dexamethasone sensitivity was defined as the ability to induce a significant dose–response change from unstimulated control samples [35,54]. The effect of MO on dexamethasone sensitivity within one type of endothelial cell was determined by composite 1-way ANOVA followed by post hoc LSD analysis. Differences between the four groups: CO/MO ShUVECs and ShUAECs in both basal and dexamethasone treatments were determined by 2-way ANOVA followed by post hoc LSD analysis. Levene’s test was used to determine variance homogeneity. Non-parametric data were analyzed using Kruskal–Wallis tests. A p-value of less than 0.05 was regarded as significant. All statistical analyses were performed using SPSS, version 26.

## Figures and Tables

**Figure 1 ijms-25-02311-f001:**
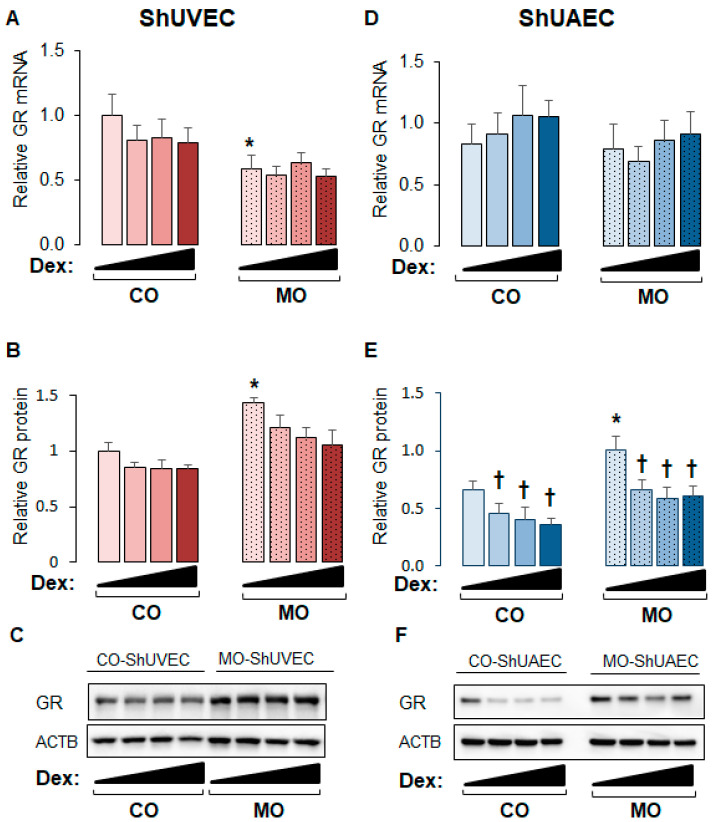
Maternal obesity upregulates GR protein expression in fetal umbilical vein and artery endothelial cells. Confluent and quiescent cells were treated with various doses of dexamethasone (0.04, 0.2, and 1 µM) for 24h and then analyzed for GR expression as described in the Section 4. (**A**) ShUVEC GR mRNA levels, (**B**) ShUVEC GR protein levels, (**C**) representative immunoblots for ShUVEC GR and ACTB, (**D**) ShUAEC GR mRNA levels, (**E**) ShUAEC GR protein levels, and (**F**) representative immunoblots for ShUAEC GR and ACTB. Bar graphs represent the mean ± SEM for each group (n = 5–6 lambs/diet group). * *p* < 0.05 control (CO) vs. obesogenic (MO) diet; ^†^
*p* < 0.05 basal vs. dexamethasone.

**Figure 2 ijms-25-02311-f002:**
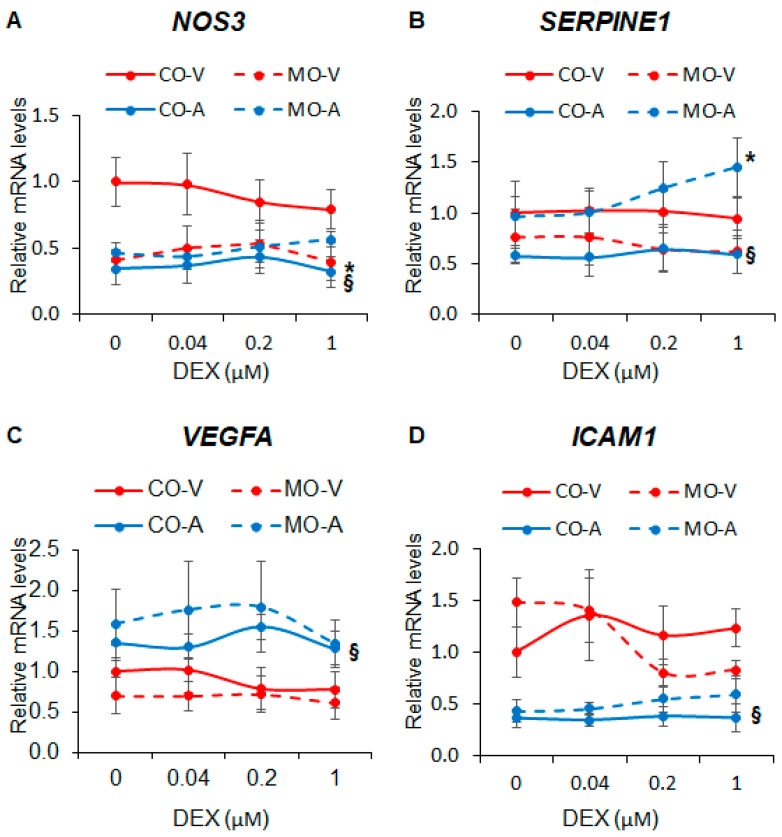
Maternal obesity alters basal fetal endothelial cell gene expression but does not affect dexamethasone response. Confluent and quiescent ShUVECs and ShUAECs were treated with various doses of dexamethasone (0.04, 0.2, and 1 µM) for 24 h and then analyzed for NOS3 (**A**), SERPINE1 (**B**), VEGFA (**C**), and ICAM1 (**D**) expression as described in the Section 4. A 2-way ANOVA followed by post hoc analysis was used to determine differences between the four groups (CO-A: control ShUAEC, MO-A: Obese ShUAEC, CO-V: control ShUVEC, and MO-V: obese ShUVEC, n = 5–6/group). * *p* < 0.05 control vs. obesogenic diet for each type of cell, ^§^
*p* < 0.05 ShUAECs vs. ShUVECs within the same diet group.

**Figure 3 ijms-25-02311-f003:**
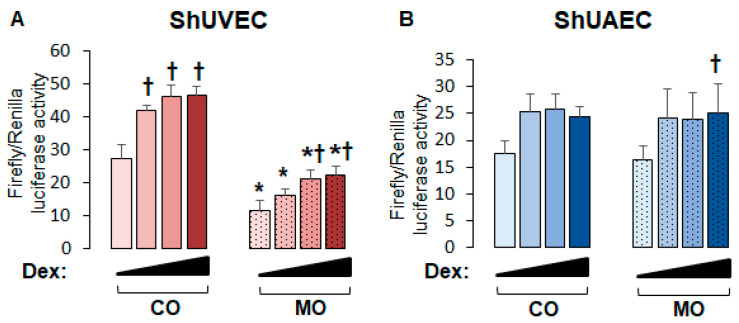
Maternal obesity decreases GRE-dependent transcriptional activation in ShUVECs. Cells were transfected with a GRE-dependent luciferase reporter vector and treated with various doses of dexamethasone (0.04, 0.2, and 1 µM) to stimulate luciferase expression. Results are shown for basal and dexamethasone-stimulated GRE-mediated transactivation in ShUVECs (**A**) and ShUAECs (**B**). Bar graphs represent the mean ± SEM for each group (n = 4–5 lambs/diet group). * *p* < 0.05 control (CO) vs. obesogenic (MO) diet; ^†^
*p* < 0.05 basal vs. dexamethasone.

**Figure 4 ijms-25-02311-f004:**
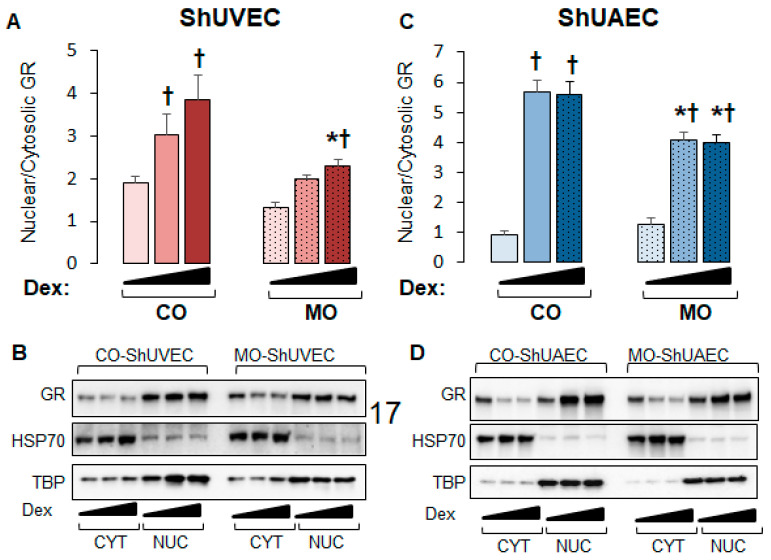
Maternal obesity decreases GR nuclear translocation in ShUVECs and ShUAECs. Confluent and quiescent cells were treated with solvent (DMSO) or dexamethasone at low (0.1 µM) and high (1 µM) concentrations for 15 min to study GR nuclear translocation as explained in the Section 4. Nuclear-to-cytosolic GR ratios and representative immunoblots are shown for ShUVECs (**A**,**B**) and ShUAECs (**C**,**D**). Bar graphs represent the mean ± SEM (n = 4–5/group). * *p* < 0.05 control (CO) vs. obesogenic (MO) diet; ^†^
*p* < 0.05 basal vs. dexamethasone.

**Figure 5 ijms-25-02311-f005:**
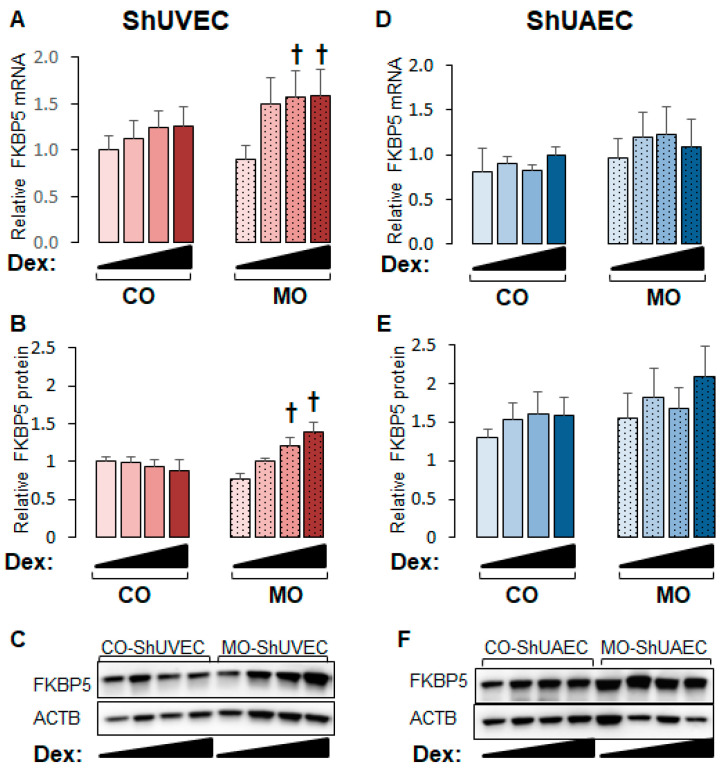
Maternal obesity increases dexamethasone sensitivity to FKBP5 upregulation in ShUVECs. Confluent and quiescent ShUVECs and ShUAECs were treated with various doses of dexamethasone (0.04, 0.2, and 1 µM) for 24h and then analyzed for FKBP5 expression as described in the Section 4. (**A**) ShUVEC FKBP5 mRNA levels, (**B**) ShUVEC FKBP5 protein levels, (**C**) representative immunoblots for ShUVEC FKBP5 and ACTB, (**D**) ShUAEC FKBP5 mRNA levels, (**E**) ShUAEC FKBP5 protein levels, and (**F**) representative immunoblots for ShUAEC FKBP5 and ACTB. Bar graphs represent the mean ± SEM for each group (n = 5–6 lambs/diet group). *p* < 0.05 control (CO) vs. obesogenic (MO) diet, ^†^
*p* < 0.05 basal vs. dexamethasone.

**Figure 6 ijms-25-02311-f006:**
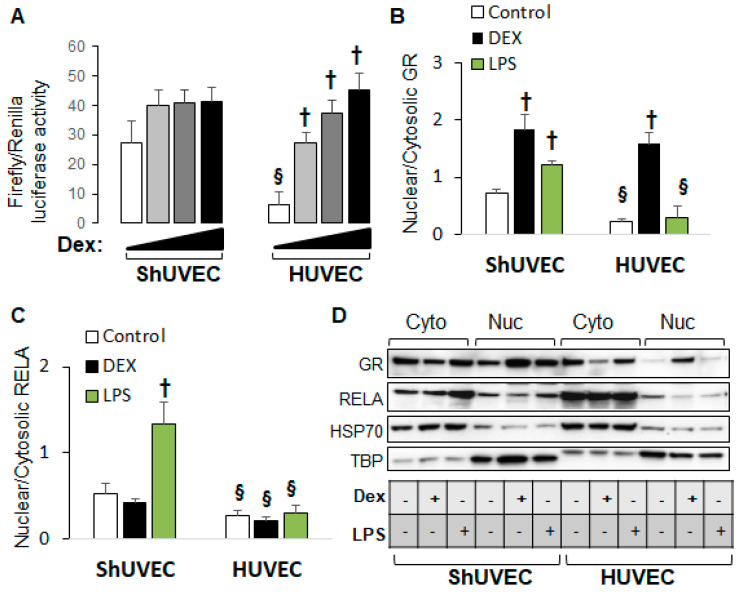
ShUVECs show decreased sensitivity to glucocorticoids compared with HUVECs. ShUVECs from control lambs of another sheep breed (Western Mix) were studied next to previously characterized dexamethasone-sensitive HUVECs. (**A**) GRE-dependent transactivation of a luciferase reporter vector is shown as relative Firefly/Renilla luciferase activity (n = 6 WM-ShUVECs, n = 12 HUVECs). (**B**) GR and (**C**) RELA nuclear translocation were studied by immunoblotting using HSP70 and TBP as control markers (**D**) (n = 4/group). ^†^
*p* < 0.05 basal vs. dexamethasone, ^§^
*p* < 0.05 WM-ShUVECs vs. HUVECs.

**Figure 7 ijms-25-02311-f007:**
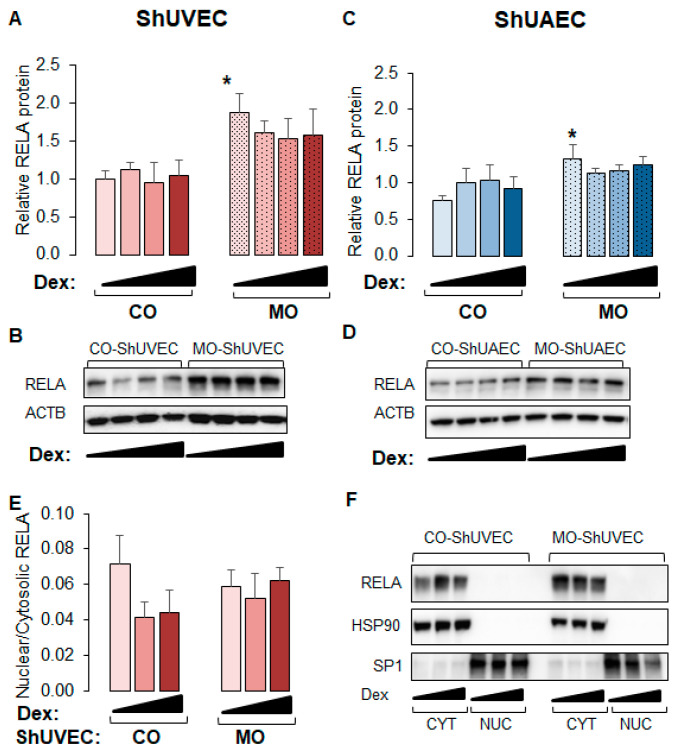
Maternal obesity increases p65 NF-kB (RELA) levels in ShUVECs and ShUAECs. Confluent and quiescent cells were treated with solvent (DMSO) or dexamethasone (Dex) to examine RELA expression by immunoblotting as explained in the Section 4. (**A**,**B**) ShUVEC total basal and dexamethasone-stimulated levels of RELA (**A**) and representative immunoblots (**B**). (**C**,**D**) ShUAEC total basal and dexamethasone-stimulated RELA expression (**C**) and representative immunoblots (**D**). (**E**,**F**) Nuclear-to-cytosolic RELA ratios (**E**) and representative immunoblots (**F**) are shown for ShUVECs. Bar graphs represent the mean ± SEM (n = 4–5/group). * *p* < 0.05 control (CO) vs. obesogenic (MO) diet.

**Figure 8 ijms-25-02311-f008:**
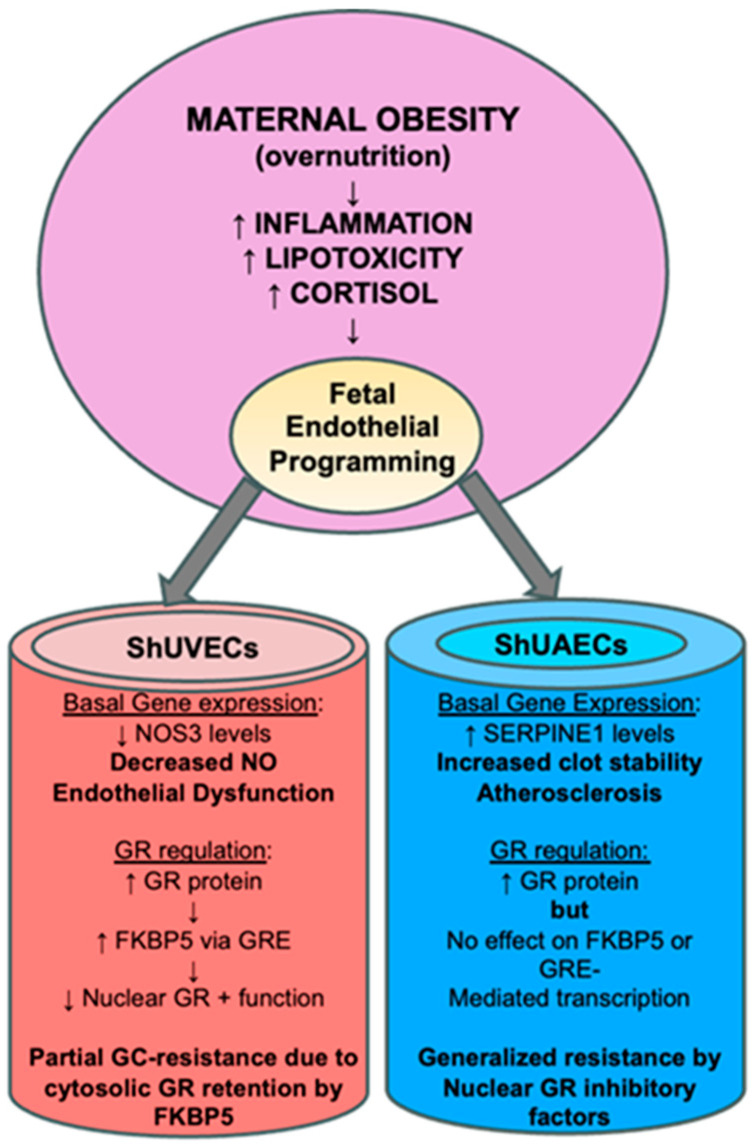
Summarized findings of MO effects on GR homeostasis and key endothelial gene expression. The current hypothesis is that MO-originated inflammatory lipid mediators induce epigenetic changes that lead to offspring endothelial dysfunction. In this MO model, fetal endothelial cells show partial to generalized glucocorticoid resistance that could be a species-specific protective mechanism against cortisol-mediated endothelial dysfunction.

**Table 1 ijms-25-02311-t001:** Maternal and fetal/neonatal characteristics of the study groups.

Characteristics ^a^	Control Diet (n = 6)	Obesogenic DIET (n = 6)	*p*-Value
Maternal			
Age (yr)	4.8 ± 0.5	4.2 ± 0.6	0.405
Pre-pregnancy weight (kg) Necropsy weight (kg) Weight gain (kg)	65.8 ± 3.9 69.3 ± 3.7 3.6 ± 1.9	69.3 ± 3.7 109.6 ± 5.2 38.2 ± 3.6	0.294 <0.001 <0.001
Fetal			
Gestational age (days)	138.5 ± 0.5	139.5 ± 0.6	0.214
Sex (male/female)	3/3	3/3	1.00
Birthweight (kg)	5.53 ± 0.27	6.06 ± 0.18	0.128

^a^ Characteristics with continuous data are shown as mean ± standard error. The statistical significance of continuous data was determined by the Student’s *t*-test. Categorical characteristics were analyzed by Pearson’s Chi-square.

**Table 2 ijms-25-02311-t002:** Antibodies used for immunoblotting.

Name	Vendor	Catalog	Dilution	Observed MW (kDa)
Glucocorticoid receptor (GR/NR3C1)	Cell Signaling Technology	12041	1:1000	95/90
FKBP prolyl isomerase 5 (FKBP5)	Cell Signaling Technology	12210	1:1000	55
NFKB p65 subunit (RELA)	Cell Signaling Technology	8242	1:1000	65
TATA binding protein (TBP)	Proteintech	22006-1-AP	1:1000	38
Specificity protein 1 (SP1)	Proteintech	21962-1-AP	1:1000	95
Heat shock protein 70 (HSP70)	Cell Signaling Technology	4872	1:1000	70
Heat shock protein 90 (HSP90)	Proteintech	13171-1-AP	1:2000	90
Beta-actin (ACTB)	Sigma Millipore	A5441	1:2000	41

**Table 3 ijms-25-02311-t003:** Primers for real-time SYBR green PCR.

Name	Sequence	Reference
Ovine glucocorticoid receptor (*NR3C1*)	Forward: 5’-CAGTGAAATGGGCAAAGGCAA-3’ Reverse: 5’-TGCGCTTGACTGTCTGTATGA-3’	NM_001114186.1
Ovine endothelial nitric oxide synthase (*NOS3*)	Forward: 5’-TCTTCCACCAGGAGATGGTC-3’ Reverse: 5’-AGAGGCGTACAGGATGGTTG-3’	NM_001129901.1
Ovine plasminogen activator inhibitor 1 (*SERPINE1*)	Forward: 5’-GACCGCAACGTGGTTTTCTC-3’ Reverse: 5’-CGAGCTCCTTGTACAGTCGG-3’	NM_001174114.2
Ovine intercellular adhesion molecule 1 (*ICAM1*)	Forward: 5’-GGTCTGGAGGTGCCGAAATA-3’ Reverse: 5’-CGGCTCACTTCCTCCTTGTT-3’	NM_001009731.1
Ovine vascular endothelial growth factor (*VEGFA*)	Forward: 5’-AAGAAAATCCCTGTGGGCCT-3’ Reverse: 5-GGAACATTTACACGTCTGCGG-3’	AF071015.1
Ovine FKBP prolyl isomerase 5 (*FKBP5*)	Forward: 5’-GAGCAGGACGCAAAGGAAGA-3’ Reverse: 5’-TGGTGTCATACGTGCCCTTC-3’	XM_042237053.2
Ovine beta-actin (*ACTB*)	Forward: 5’-GCAGATGTGGATCAGCAAGC-3’ Reverse: 5’-GGGTGTAACGCAGCTAACAG-3’	NM_001009784.3
Ovine 18S ribosomal RNA (*18S* rRNA)	Forward, 5’-AAACGGCTACCACATCCAAG -3’ Reverse, 5’-TCCTGTATTGTTATTTTTCGTCAC-3’	AY753190.1

## Data Availability

Data are contained within the article.

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
