# Peer review of "Dysregulation of Glucocorticoid Receptor Homeostasis and Glucocorticoid-Associated Genes in Umbilical Cord Endothelial Cells of Diet-Induced Obese Pregnant Sheep"

_ijms, 2024, doi:10.3390/ijms25042311_

Round 1
Reviewer 1 Report
Comments and Suggestions for Authors
The manuscript entitled "An obesogenic diet dysregulated umbilical cord endothelial cell expression and function of the glucocorticoid receptor in the pregnant sheep" by Eugenia Mata-Greenwood and colleagues addressed a very important issue in metabolic and developmental physiology and the authors substantiated the manuscript with comprehensive data sets. I have few comments which are listed below:
1. The title should be edited as "Dysregulation of glucocorticoid receptor homeostasis and glucocorticoid associated genes in the umbilical cord endothelial cells in pregnant sheep feed with obesogenic diet".
2. There are a few typographical errors. For example, Lines 36, 38, 41, 45, 46 space should be added before the reference bracket. The complete manuscript should be edited.
3. Line: 51 There is an extra space after the full stop.
4. The introduction portion should be more concise.
5. Line 103: "Our ovine model of MO" should be edited to " the ovine model of MO used in the current study.
6. The figure legends should describe A, B, .... F in figures 1-5.
7. Figure 1 (A). There is no statistical difference between CO and MO values of relative GR mRNA in ShUVECs?
8.
Author Response
General reply: Thank you for your analysis of our study. We have carefully read your concerns and have attempted to address them appropriately. Please see our answers according to each question.
- The title should be edited as "Dysregulation of glucocorticoid receptor homeostasis and glucocorticoid associated genes in the umbilical cord endothelial cells in pregnant sheep feed with obesogenic diet".
Answer: We have changed the title to: “Dysregulation of glucocorticoid receptor homeostasis and glucocorticoid associated genes in umbilical cord endothelial cells of diet-induced obese pregnant sheep”. We hope this title satisfies the reviewer.
- There are a few typographical errors. For example, Lines 36, 38, 41, 45, 46 space should be added before the reference bracket. The complete manuscript should be edited.
Answer: Thank you for pointing these grammatical errors. The entire manuscript has been corrected to have a space before the brackets with the reference numbers.
- Line: 51 There is an extra space after the full stop.
Answer: The extra space has been deleted.
- The introduction portion should be more concise.
Answer: We shortened the introduction by rearranging some key concepts. This led to the introduction being 14 lines shorter. This topic is somewhat complex and several paragraphs are needed for the introduction including: 1) MO obesity effects in offspring, 2) MO mechanisms of fetal programming, 3) glucocorticoid effects in vascular endothelial cells, 4) determination of tissue glucocorticoid sensitivity, and 5) aim of the study.
- Line 103: "Our ovine model of MO" should be edited to " the ovine model of MO used in the current study.
Answer: This sentence now reads: ‘The ovine model used in the current study’ as requested.
- The figure legends should describe A, B, .... F in figures 1-5.
Answer: We added description for all the panels for Figures 1, 3, and 5. Figures 2, 4, and 6 already had the description of all the panels. We also bolded the letters A, B….F for clarity.
- Figure 1 (A). There is no statistical difference between CO and MO values of relative GR mRNA in ShUVECs?
Answer: We repeated the qPCR again and reanalyzed the statistics using SPSS 28.0. The repeated data could be normalized by using the natural logarithm of the folds and we were able to use ANOVA analysis. As shown now in the new Figure 1, the only significant difference is that Obese-ShUVECs have lower basal levels of GR mRNA compared with Control-ShUVECs. However, dexamethasone remains without having any significant effects on GR mRNA. Since there is now significance in ShUVEC GR mRNA we now reflect that in the results section, line 94 and in the discussion section, line 264.

Reviewer 2 Report
Comments and Suggestions for Authors
An obesogenic diet dysregulates umbilical cord endothelial cell expression and function of the glucocorticoid receptor in the pregnant sheep
The study demonstrates the effect of maternal obesity on glucocorticoid receptor function. The approach and the overall design of the study are good. As inflammatory cytokines and their signaling play a central role in fetal endothelial programming, the secretion of proinflammatory cytokines needs to be validated. So estimation of cytokine levels in the conditioned medium of the treated cells should be estimated. Also, it is suggested to include the results of nuclear translocation of NFkb. Grammatical errors are noted on multiple occasions. Abbreviations need thorough revision. Expand first and then abbreviate. Abstract and manuscript sections should be self-explanatory in terms of abbreviations.
Comments on the Quality of English LanguageModerate editing of the English language is required.
Author Response
Thank you for the comments that have helped us improve this manuscript. Please see below for some detailed answers:
- “As inflammatory cytokines and their signaling play a central role in fetal endothelial programming, the secretion of proinflammatory cytokines needs to be validated. So estimation of cytokine levels in the conditioned medium of the treated cells should be estimated.”
Answer: It is well known that in non-stimulated endothelial cells the levels of IL6, TNF alpha, and other pro-inflammatory mediators are too low to be measured by ELISA. In order to study the differential activation of NF-kB in endothelial cells from control and obese pregnancies, we would have to treat the cells with a pro-inflammatory mediator such as TNF-alpha or LPS, which we are now considering as a new pilot study (see lines 360-361). However, these experiments are outside the scope of the current study which is focused on glucocorticoid receptor homeostasis.
- “Also, it is suggested to include the results of nuclear translocation of NFkb”
Answer: We are presenting additional western blots showing that MO ShUVECs and ShUAECs express higher basal levels of RELA (p65 NF-kB), which is localized mainly in the cytosol. This is shown now in Figure 7 (lines 241-249). We also added the corresponding subsection under Results (lines 232-239) and Discussion (lines 374-376 and 398). Maternal obesity increased p65 NFkB expression in both ShUVECs and ShUAECs, although without evidence of activation (as it was localized in the cytosol).
- “Grammatical errors are noted on multiple occasions.”
Answer: We checked for grammatical errors and found several in the main text and figure legends. For example, the micromolar sign was incorrectly written as uM instead of µM. We fixed all the errors that we could find.
- “Abbreviations need thorough revision. Expand first and then abbreviate. Abstract and manuscript sections should be self-explanatory in terms of abbreviations”
We checked the abbreviations and found two in the abstract that needed explanation (lines 19 and 27 of the abstract). Other than that, the other abbreviations were expanded the first time they were used.
